# Environmental Assessment of Ultra-High-Performance Concrete Using Carbon, Material, and Water Footprint

**DOI:** 10.3390/ma12060851

**Published:** 2019-03-13

**Authors:** Husam Sameer, Viktoria Weber, Clemens Mostert, Stefan Bringezu, Ekkehard Fehling, Alexander Wetzel

**Affiliations:** 1Center for Environmental Systems Research (CESR), University of Kassel, 34117 Kassel, Germany; vicky191@gmx.de (V.W.); mostert@uni-kassel.de (C.M.); bringezu@uni-kassel.de (S.B.); 2Faculty of Civil and Environmental Engineering, University of Kassel, 34125 Kassel, Germany; 3Institute of Structural Engineering (IKI), University of Kassel, 34125 Kassel, Germany; fehling@uni-kassel.de (E.F.); alexander.wetzel@uni-kassel.de (A.W.)

**Keywords:** sustainable buildings, life cycle assessment, construction materials, concrete, durability

## Abstract

There is a common understanding that the environmental impacts of construction materials should be significantly reduced. This article provides a comprehensive environmental assessment within Life Cycle Assessment (LCA) boundaries for Ultra-High-Performance Concrete (UHPC) in comparison with Conventional Concrete (CC), in terms of carbon, material, and water footprint. Environmental impacts are determined for the cradle-to-grave life cycle of the UHPC, considering precast and ready-mix concrete. The LCA shows that UHPC has higher environmental impacts per m^3^. When the functionality of UHPC is considered, at case study level, two design options of a bridge are tested, which use either totally CC (CC design) or CC enhanced with UHPC (UHPC design). The results show that the UHPC design could provide a reduction of 14%, 27%, and 43% of carbon, material, and water footprint, respectively.

## 1. Introduction

The construction industry is recognized as an issue of the global environmental impact [1]. Cement production is responsible for up to 8% of global greenhouse gas (GHG) emissions [2]. This is because of intensive use of energy for heating requirement of raw materials during manufacturing processes of cement [3,4]. Investigating efficiency of material resource use is an increasingly important area in the construction sector [5], according to the enormous increase of the construction minerals extraction [6]. Water use in the construction industry has received considerable critical attention [7] due to its intensive consumption within, for instance, cement manufacturing [8]. Investigating alternative concrete technologies is a continuing concern for defining the contribution to the environmental impact mitigation [9,10].

The innovation of Ultra-High-Performance Concrete (UHPC) is increasingly used [11,12] for its high compressive strength of more than 150 MPa with less mass relative to the compressive strength, and the highest tensile strength of more than 10 MPa [13,14,15]. UHPC, sometimes addressed as Reactive Powder Concretes (RPC), has superior technical performance, e.g., exquisite granular compactness and premium porosity reduction [16]. Compact Reinforced Composite (CRC) provides the combination of the strength with ductility, which increases the durability and formability of the concrete [17]. The experimental comparison of UHPC with Conventional Concrete (CC) indicated that the strength of the used UHPC is three to four times greater than CC strength [18]. In addition to that, UHPC has higher modulus of elasticity, higher tensile strength, and ductility. Even though UHPC is subject to an energy-intensive production process [14] and requires a relatively longer mixing time [19], it could provide a longer service life without costly maintenance measures in comparison with CC [20]. Questions have been raised regarding state of the art of UHPC research works related to its environmental performance.

However, UHPC provides a significant reduction in the ratio of mass to load-bearing capacity [14]. It is problematic to decide for instance whether material resources are efficiently used without defining life cycle wide use of material resources [5,21]. While environmental impacts of a different mixture of CC are intensively researched [22,23,24,25], substantial environmental issues, e.g., cumulative raw material used [5,26] and water use [27] are still insufficiently studied [28]. These issues have recently received major awareness [29,30,31,32,33]. There is a general lack of studies on LCA of UHPC, Stengel and Schießl [34] investigated environmental impact of three bridges which essentially erected using UHPC in three different countries Canada, Germany, and USA. Randl et al. [35] investigated environmental impacts of concrete using efficient energy hydraulic additives instead of cement in UHPC. Consequently, regarding the presented studies, global warming impact is considered, whereas life cycle wide use of material and water resource are still missing.

Environmental performance of construction materials could be comprehensively described by using LCA [36] based on ISO 14040 and ISO 14044 [37]. Sustainability assessment of natural resources, e.g., water and material resources, is a key aspect of achieving the Sustainability Development Goals (SDG) [38]. Recently, indicators of material resource use, i.e., Raw Material Input (RMI) and Total Material Requirement (TMR), which are within the focus of this article, have been defined as crucial indicators to be considered within sustainability assessment schemes of buildings [5]. In addition, it is highly recommended to use the AWARE (Assessing impacts of Water consumption based on Available Water Remaining) method for the assessment of water footprint [27].

This article systematically investigates carbon, material, and water footprint of UHPC in comparison with CC, aiming to facilitate the decision-making process, undertaken by the construction planners. The footprints are assessed within cradle-to-grave life cycle stages according to EN 15804 [32].

## 2. Materials and Methods

### 2.1. Reference Mixtures of Concrete

Specifications of concrete are regulated in DIN EN 206 [39]. Two mixtures of UHPC, M3Q and M2Q, are selected from the results of a research project within the German priority program 1182 [14], for the ready-mix concrete, M3Q mixture of UHPC is considered and for the precast concrete, M2Q mixture of UHPC is considered. The environmental analysis is done in comparison with two mixtures of CC, ready-mix concrete (C35/45) and precast concrete (C50/60) [40], where 35 and 50 refer to the minimum compressive strength in MPa of a cylinder specimen of concrete with 150 mm in diameter and 300 mm in length, respectively, and 45 and 60 refer to the minimum compressive strength in MPa of a cube specimen of concrete with 150 mm edge length after 28 days when tested according to DIN EN 206 [39]. Mixtures of CC and UHPC are shown in Table 1.

UHPC is known for its high compressive strength of more than 150 MPa with low water/cement ratio [11]. Schmidt et al. [14] defined the type of cement for M3Q and M2Q as fast hardening Portland cement (CEM I 52.5) with high sulfate resistance and low alkali content of 52.5, which represents the minimum compressive strength at 28 days according to EN 197-1 [41].

For the CC mixtures, the average mixture of concrete in Germany is used as it has been described for the environmental product declarations (EPD) of the concrete by the German Federal Association of Concrete [40]. For the comparison’s purposes, quantities of aggregates are defined as primary raw materials by considering the proportion of the recycled aggregate within the quantity of primary aggregate. In order to define type of cement, average content of clinker in cement is 85% [42], which corresponds to the cement type of CEM II/A according to EN 197-1 [41].

### 2.2. Life Cycle Assessment

The Life Cycle Inventory (LCI) of UHPC in comparison with CC is described based on the Life Cycle Stages according to EN 15804 [32]. LCI Input and output environmental flows of are shown in Figure 1. The database of GaBi XIV construction materials [43] is used with openLCA software [44] for the LCA modeling for all processes except the production process of steel fibers, which is not available within the GaBi XIV database. Therefore, ecoinvent 3.1 database [45] is used for the LCA modeling of steel fibers. In terms of the comparison at the construction materials level, LCI is described using average values in Germany. Interviews with stakeholders have been conducted for the non-documented data such as transportation distance of quartz sand to the concrete plant. LCI processes are described in Appendix A, Appendix B and Appendix C. Secondary materials are allocated according to EN 15804 [32]. Fly ash is a by-product of the coal power station. Silica fume is a by-product of the production of silicon metal or ferrosilicon alloys. Therefore, their environmental impacts are only considered for the transport to the concrete plant.

#### 2.2.1. Life Cycle Stages A1–A3

A1 life cycle stage of the concrete represents cradle-to-grave inventory (A1–A3) of its ingredient, for instance, A1–A3 production process of cement. Cement production includes raw material extraction (A1), transport of the raw materials (A2), and production process (A3), e.g., manufacturing of clinker, and cement grinding. Production process of quartz sand and quartz powder is defined within GaBi XIV construction materials [43]. This process includes grinding into smaller particle size and separation from impurities. Superplasticizers are defined according to the environmental product declaration (EPD) of the German Association Deutsche Bauchemie [46]. Production process of steel fibers is exemplified by Stengel and Schießl [47]. The process chain includes crude steel production, hot rolling to wire, descaling, dry drawing, wet drawing, annealing, twisting, and cutting to length.

A2 life cycle stage represents the transport of concrete ingredients to the concrete plant. Transporting types and distances of cement, aggregate, fly ash, plasticizers are taken from an LCA study conducted by the German ready-mixed concrete association [48], while interviews are undertaken with experts from stakeholder for getting transporting information of quartz sand, quartz powder, silica fume and steel fibers. Types of transport and distances are shown in Table 2.

A3 life cycle stage comprises the concrete mixing process. Concerning CC, life cycle inventories of electricity, diesel and fuel oil demand are analyzed according to data from a technical report of energetic optimization of concrete production of the ready-mix concrete plant done at the University of Stuttgart [49]. The electricity, diesel and fuel oil demand are determined by averaging the energy demand of three concrete plants that were analyzed in in Reference [49]. It contains, in addition to the single mixing process, the whole needs of the concrete plant, for instance, lighting and laboratory operation requirements. The energy demand to produce UHPC has to be calculated according to the longer mixing process of 10–15 min [19]. The ready-mix concrete is compacted at the construction site, which refers to A5 life cycle stage.

In addition, A3 life cycle stage of precast concrete includes compacting and heating treatment of the concrete. The research project of the Technical University of Cottbus [50] describes the electricity demand of two high-frequency internal vibrators by considering a power of 1.5 kW and a compacting period of five minutes per cubic meter with a utilization rate of 100%; the electricity demand is 0.25 kWh/m^3^. The modeling of the heat treatment is based on the expert’s interviews. Therefore, an amount of 3 L/m^3^ of fuel oil is required. Table 3 shows the need for electricity, fuel oil and diesel for the A3 life cycle of the production of 1 m^3^ concrete.

#### 2.2.2. Life Cycle Stages A4–A5

LCI of the A4 life cycle stage of CC and UHPC are the same. The ready-mix concrete is transported by a truck mixer from the concrete plant to the construction site. The GaBi database [43] does not contain such a process. Therefore, the transport of the ready-mix concrete is modeled by using a conventional truck and adding an additional diesel input regarding the higher fuel consumption of a truck mixer. The average fuel consumption of a truck mixer is 0.48 L/km [51]. The truck has a fuel consumption of 0.35 L/km [43]. Consequently, additional input of 0.13 L/km is added. Data of the average transport distance is based on the annual report in 2017 of the German Federation of the ready-mixed concrete [52], which is 16.3 km as the average transport distance for ready-mix concrete 2014–2016.

The transport of the precast concrete is modeled by using a truck with a total weight of 34–40 t and a capacity of 27 t with fuel consumption of 0.46 L/km [43]. Environmental product declaration of structural concrete [40] defined an average distance of 180 km for the transporting of the precast concrete. The utilization rate of the trucks capacity was assumed to be 100% on the way to the construction site and 0% on the return which leads to an average rate of 50%, life cycle stage, A5, includes installation at the construction site. Concrete truck pump and compactors are using for the installation of ready-mix concrete. The technical data are specified according to the research project of the Technical University of Cottbus [50]. Power of concrete truck pump is 231 kW and an output capacity of 164 m^3^/h, with a utilization rate of 50%. This is equivalent to an operating period of 0.012 h/m^3^. Precast concrete is either installed by using a revolving tower crane or by using a mobile crane [40]. In terms of the considered case study (see Section 2.4), a mobile crane is selected with a maximum payload of 80 t and a power of 320 kW [50].

#### 2.2.3. Life Cycle Stages C1–C3

C1 life cycle stage of UHPC and CC refers to the deconstruction measures. They differ from each other in the type, the capacity, and the fuel consumption of excavator. CC could be demolished using an excavator with a total weight of 30 t and fuel consumption of 18 L/h [50]. The capacity of the machine is assumed to be 30 m^3^/h according to the recommendations from experts. In terms of UHPC, there is no documented practical experience concerning the demolishing process [14]. A more powerful machine and a longer deconstruction period are recommended by the experts; therefore, the diesel input is determined of an excavator with a total weight of 50 t, fuel consumption of 35 L/h and a capacity of 5 m^3^/h.

Transport to the recycling plant represents C2 life cycle stage. In Germany, the construction waste is usually transported by trucks with a total weight of 40 t [53]. The distance to the recycling plant is 30 km [50]. The utilization rate of the trucks is assumed to be 50%.

Life cycle stage of C3 includes the treatment of the construction waste. The waste of CC is going to be pre-crushed with an excavator and transported to the processing plant by a wheel loader, related technical specifications are described by Heyn and Mettke [50]. In case of UHPC, the same specifications of the wheel loader could be considered. For the pre-crushing using the excavator, the experts recommended that an additional utilization rate of 20% should be considered in comparison with the treatment of CC. Related LCI processes are described within the SI 4.

### 2.3. Footprint Categories

#### 2.3.1. Carbon Footprint

The Joint Research Center (JRC) of the European Commission recommends using characterization factors (CFs) of a 100-year time horizon global warming potential (GWP_100_) for defining the carbon footprint [54]. The carbon footprint refers to the GHG emission accompanied within the life cycle process chain of a product quantified in kg CO_2_ equivalents. CFs are documented in the fifth assessment report (AR5) by the Intergovernmental Panel on Climate Change (IPCC) [55]. The elementary flows are set up to be comparable with GaBi construction material database in openLCA software [5].

#### 2.3.2. Material Footprint

Material footprint is defined in terms of Raw Material Input (RMI) and Total Material Requirement (TMR). RMI measures cumulative raw materials used within the whole process chain of a product [26,56]. Used and unused extractions could be determined by TMR. The unused extraction refers to all material that is taken from the environment to enable the extraction of the primary raw material [57].

The TMR thus measures the total amount of abiotic and biotic primary material required over the complete life cycle [21,58]. Regarding construction materials, RMI and TMR address all materials input from the environment during production, transport, use, maintenance, demolishing, and recycling [5].

CFs of RMI are developed by the Sustainable Resource Futures (SURF) group in the Center for Environmental Systems Research (CESR) at the University of Kassel [57,59,60], whereas the background methodology is well described and is applied for the assessment of the resource use of the building’s exterior walls [5]. TMR measures the total of the RMI plus unused extractions [58]. Data on unused extractions is documented by WU [61].

#### 2.3.3. Water Footprint

Water footprint is assessed according to the AWARE method [27], which is the current WULCA (Water Use in LCA) consensus LCA method for water scarcity in compliance with ISO 14046 [62]. Considering hydrological water availability, human water consumption and needs of aquatic freshwater ecosystem, impacts of water use are expressed in terms of the indicator available water remaining per water shed and country.

AWARE aims at identifying the risk of freshwater depletion for any other water user in the same area. The resulting indicator Availability-Minus-Demand with the unit m^3^/m^2^ * month of any basin is related to the world average to calculate regionalized CFs, which are available online http://www.wulca-waterlca.org/aware.html.

The CFs represent “a relative value of the impact score of water consumption” [27], (p. 374) in comparison with the world average consumed in a basin. CFs are ranged from 0.1 to 100 m^3^ world eq/m^3^. For instance, the German CF of 1.36 [27] indicates that in Germany 1.36 times less water is remaining in a specific period than in the world average, whereas the world average CF is equal to 42.95 [27]. Regions with less than one percent of the world average available water remaining are automatically set to the maximum value of 100, thus being marked as extreme arid areas.

Water consumption in different regions is comparable due to considering the units m^2^·month equivalent in every basin: Consuming water in two regions with the same available water remaining per m^2^·month is therefore assumed to be equal. Concerning the LCA calculations related to Section 3 of this article, both products are produced in Germany, therefore water consumption is weighted using German AWARE CFs. For the purpose of transferring the methodology to processes not located in Germany, the CFs must be adapted accordingly.

### 2.4. Case Study Description

Two design options of an overpass road bridge in Germany (Obertiefenbach Bridge) are tested [63,64]. The first design option implied the exclusive use of CC (CC-Design) and the second one is a combination of CC and UHPC (UHPC Design). The bridge should provide a shorter time of construction and high performance in terms of durability, resistance to the mechanical loads, and stiffness. Therefore, UHPC is suggested to enhance the design of the bridge. UHPC was proposed to substitute use of the CC in the top layer of the deck slab, hinge over the middle piers, and the kerb. All the other elements of the bridge, e.g., main girders, abutments, and middle piers are designed to be built using CC. Construction materials are shown in Table 4. For more information regarding drawings and material specifications, see Brühwiler et al. [63] and Schmidt et al. [64].

However, the design of the bridge is wholly described by Brühwiler et al. [63]. An overview of UHPC Design will be shortly discussed. The bridge consists of two-span, 22.5 m and 21.5 m, with main five girders supported by the abutments and one middle support. The middle support consists of five single piers to support the five girders without a cross girder to provide efficiency in lighting. The main girders are precast, prestress, and longitudinally pretension reinforced concrete. When the main girders are installed on the abutments from the ends and piers from the middle, in the long direction, the space between them in the middle is filled out with UHPC. Then a layer of UHPC (3 cm) is cast on the top surface of the deck slab for the connection of the girders and waterproofing protection. Kerbs are prefabricated from UHPC, then glued on the top surface of the bridge.

The system boundaries include the whole life cycle from the raw material extraction to the end-of-life phase of the bridge within the two design options. LCA calculations are done within the GaBi database. Regarding materials, which are not available in the GaBi database, i.e., steel fibers, they are modeled within the ecoinvent database [45]. For the sealing layer, a standard bituminous sheeting with a polyester fleece bearing is used. Below the kerb elements, a glass fleece bituminous sheeting is applied. LCA modeling of A1–A3 is typical to Section 2.2.1.

In terms of the module A4, for the transportation modeling, van, truck, and road train are used for the transportation modeling. The choices of the transport type depend on the amount of transported material. Transportation distances are defined according to Lünser [65]. An additional amount of diesel should be added to the transport process of the concrete regarding Section 2.2.2. During the transport of poured asphalt, the boiler needs an additional input of diesel and fuel oil. According to the expert interviews, the diesel demand of a boiler is 8 L/h and the fuel demand is 3.5 L/h.

A5 represents the installation of precast and ready-mix concrete at the construction site. Diesel demand of the construction equipment and the amount of liquefied petroleum gas to install the bituminous asphalt sheeting are defined according to the expert’s interviews.

The lifespan of the bridge, use phase B4, is assumed to be 90 years [65]. Maintenance works and lifespan of the structure elements bridge are modeled according to Lünser [65] as 20 years for the lifespan of the road surface and sealing, and 30 years for the lifespan of kerb elements and concrete surface. LCA calculations contain all expenditures for the removal and renewal (replacement) of construction materials including the transport to and from the construction site. A number of the replacements are shown in Table 5. Road surface and sealing must be renewed four times. Kerb elements and concrete surface must be renewed two times. The concrete surface refers to the part beneath the sealing layer which is often damaged by deicing salts. It is assumed that after 30 years 20% of the area would need to be renewed. In case of the UHPC Design, the sealing layer is not existent. Instead, it has a 3 cm thick UHPC layer which prevents a possible deicing attack due to a very high impermeability [66]. The high durability of UHPC facilitates a long lifetime and does not need to be renewed after 30 years. Nevertheless, it is assumed that the asphalt overlay would be renewed after 20 years. Furthermore, the UHPC kerb elements would not need to be removed after 30 years. The module B4 is divided into B4 (a): renewal of road surface and sealing and B4 (b): renewal of kerb elements and concrete surface.

For the deconstruction life cycle stage, C1, crowbar, excavator loader, hydraulic hammer, wheel loader, and cutters are assumed to be used; They require a diesel consumption of 17.6 L/m^3^ in terms of CC [65]. To consider the higher performance of UHPC the fuel consumption is doubled resulting in 35.2 L/m^3^ according to the experts’ interviews. A milling machine with 2.54 L/m^3^ diesel demand could be used for the deconstruction of the road surface, and the deconstruction of the sealing could be done using an excavator with 0.20 L/m^2^ diesel demand and 0.13 L/m^2^ liquefied petroleum gas demand [65]. The construction waste is sorted by material and transported, C2 life cycle stage. The transportation is assumed to be done by truck (27 t payload) over a distance of 30 km) to the recycling plant [50]. Life cycle stage of C3 represents treatment of the construction waste. LCA is modeled for the needed wheel loader, excavator, and other processes in relation to waste plant. In terms of CC, the processes needed are a wheel loader with 0.60 L/m^3^ diesel demand, excavator with 0.30 L/m^3^ electricity demand of 18.26 L/m^3^ for other processes within a waste plant [65]. Regarding UHPC, however, the same demand of diesel of the wheel loader is needed, and an additional diesel demand of excavator is needed, i.e., 0.43 L/m^3^ and simultaneously 21.21 MJ/t of electricity is needed for other processes within the waste plant.

### 2.5. Definition of The Functional Unit

ISO 14040 [67] defined the functional unit of a product system as a measure of the benefit of the functional outputs. However, the environmental footprints of the UHPC in comparison with CC should be provided within the functionality of the final products, e.g., bridges, because UHPC could strongly reduce mass to load-bearing capacity. LCA results for 1 m^3^ of UHPC in comparison with CC will be discussed in Section 3.1 and Section 3.2 in order to enhance future studies with other applications. The results related to a practical application of UHPC will be shown within the Section 3.3. When analyzing and comparing the life cycle of bridges, a functional unit should represent an equal practical value [65]. LCA results are done for the two design variants of the bridge. The practical value (functional unit) is defined as the whole bridge because two design options are connecting two identical places, assigned to the same bridge class, and having the same load capacity and length.

## 3. Results and Discussion

### 3.1. Construction Materials Level

#### 3.1.1. Carbon Footprint at Construction Materials Level

LCA results of the carbon footprint of 1 m^3^ of UHPC in comparison with 1 m^3^ of CC are shown in Figure 2. The results show a higher quantity of GHG of UHPC. A1–A3 life cycle stages contribute to more than 90% of the carbon footprint of all the considered mixtures of concrete. Therefore, the share contributions of the materials within A1–A3 are additionally discussed. Regarding UHPC, GHG emissions of the production of the steel fibers are contributing to more than 40% of the carbon footprint of M3Q and M2Q. Cement production is responsible for more than 80% of the GHG emissions of the considered mixtures of CC, C35/C45 and C50/60.

#### 3.1.2. Material Footprint at Construction Materials Level

LCA results of the material footprint (measured in RMI and TMR) are shown in Figure 3. The material footprint of UHPC is much higher than that one of the CC. Figure 3 reveals that there has been a marked increase in the values of TMR of UHPC mixtures, i.e., M3Q and M2Q, owing to high value of unused per used extractions of the production of the steel fibers. In terms of life cycle stages A1–A3 of the CC (C35/45 and C50/60), material footprint is mainly dominated by more than 65% from the production process of aggregate and by more than 30% from the production process of cement.

#### 3.1.3. Water Footprint at Construction Materials Level

Figure 4 shows that the water footprint is three times increased when UHPC is used. The largest volume of water is consumed by the M2Q mixture of concrete. A1–A3 life cycle stages are majorly responsible for the water footprint in comparison with other life cycle stages. The production process of cement dominates the water footprint. Due to the higher cement content in the ultra-high-strength concretes, its water footprint is correspondingly high.

### 3.2. Sensitivity Analysis at The Construction Materials Level

The evaluation of the considered footprints shows that the UHPC (M2Q) has the highest environmental impact. The analysis results of UHPC ready-mix concrete (M3Q) show slightly lower environmental impacts but they are still higher than CC results. About CC, the concrete of compressive strength class C50/60 has more environmental impact. Furthermore, as a commonality of the three footprints can be seen that the highest environmental impact occurs in the production stage A1–A3. These contributed to more than 95% to the material footprint and 90% to the carbon and water footprint. Dominant processes of the carbon and water footprint are the production of cement and steel fibers, as well as the aggregates of the material footprint. The micro steel fibers have been also identified for the largest unused per used extractions.

The sensitivity analysis will examine how the footprints are sensitive to changing the two dominants variants as follows:Type of cement.Production process of micro steel fibers.

#### 3.2.1. Type of Cement

The CEM I 52.5 R-HS/NA is used as the type of cement for the UHPC [14]. This is a fast hardening Portland cement with a high sulfate resistance (HS) and a low alkali content (NA). The Portland cement CEM I has high clinker content of 95–100% according to EN 197-1 [41]. High clinker content of the cement increases the accompanied GHG emissions, whereas, cement types CEM II–CEM V have a lower clinker content. The UHPC practice applications listed in the literature have been mostly performed with C3A lean Portland cement. However, in the Netherlands, a bridge of UHPC was built with a CEM III 52.5 [68]. According to EN 197-1 [41], the type of cement CEM III/A has a clinker content of 35–64%. The sensitivity analysis will include using of cement type CEM III/A as a binder for all mixtures of concrete as shown in Table 6.

#### 3.2.2. Production Process of Steel Fibers

The main part of steel production for instance in Germany takes place in the blast furnace route [69]. Pig iron is made in the blast furnace route using coke; consequently, the iron is then converted to steel in the basic oxygen furnace. 20–30% of scrap is used within blast furnace route, whereas more than 42% of scrap is used within the production of steel in an electric arc furnace with less energy use [69]. Therefore, the sensitivity to the environmental impacts of changing the type of the production of the steel fibers of UHPC will be examined.

#### 3.2.3. Results of The Sensitivity Analysis

Sensitivity of the footprints to the changing type of cement for all mixtures of UHPC and CC and type of production process of steel fibers for UHPC mixture will be shown in this section. Figure 5 shows that 42% of the carbon footprint of the UHPC mixtures is decreased. Potential saving of more than 30% of the carbon footprint of CC is accordingly presented.

Regarding material footprint, the sensitivity analysis indicates that a potential saving of 62% could be done within the UHPC mixtures as shown in Figure 6. This is mainly a sequence of using a high amount of secondary materials. TMR of CC is decreased by up to 17%.

Figure 7 shows that the water footprint could be slightly affected in comparison to other considered footprints. Using CEM III 52.5 as the type of cement for all mixtures of concrete and an electric arc furnace for steel fibers production could provide up to 17% less water footprint of UHPC mixtures and approximately 3% less water footprint of CC mixtures.

### 3.3. Case Study Level

#### 3.3.1. Carbon Footprint at Case Study Level

The carbon footprint of the bridge with CC (CC-Design) could be decreased by about 40 t CO_2_ eq. when the UHPC is considered to enhance the design of the bridge (UHPC Design) as shown in Figure 8. In terms of CC-Design, 63% of the carbon footprint comes from A1–A3 life cycle stages, whereas those stages contribute to 92% of the carbon footprint of the bridge when the UHPC is used. Analysis of A1–A3 stages shows that production processes of CC and steel mainly contribute to the carbon footprint of the CC-Design. On the other hand, production processes of UHPC, CC, and steel dominate GHG emissions of the bridge when UHPC is used altogether with CC (UHPC Design).

#### 3.3.2. Material Footprint at Case Study Level

Figure 9 shows that a smaller material footprint could be seen when the UHPC Design is considered with a potential saving of more than 8 t of the cumulative raw material used (RMI) and more than 6.5 t in terms of TMR. CC and steel production processes contribute to 80% and 13% respectively to the material footprint of CC-Design. In terms of UHPC Design, CC has also the highest impact on the material footprint with 47%, then comes the UHPC with 38%.

### 3.4. Water Footprint at Case Study Level

Water footprint could be reduced by more than 30% when the UHPC design is considered, as shown in Figure 10. Regarding the two design options of the bridge, production process of the steel is mainly responsible for the water footprint of 64% for the CC design and 59% for the UHPC design.

### 3.5. Summary of Analyses at Material and Case Study Level

In order to assess the impact of using UHPC, comparisons on material level are not sufficient. Hence, the functional unit should be considered at the case study level. Therefore, the environmental footprints of the UHPC is tested in comparison with CC on a practical application (design of an overpass road bridge). In contrast to the results for the environmental footprint of 1 m^3^ of concrete mixtures (Section 3.1 and Section 3.2), the analysis results of the bridge design show that the design enhanced with UHPC (UHPC Design) could provide better environmental performance in terms of carbon, material, and water footprint. Results show that the environmental footprint is dominated by the A1–A3 life cycle stage. Therefore, those stages are analyzed more closely.

## 4. Conclusions

This article provides a comprehensive LCA of carbon, material, and water footprint of UHPC in comparison with CC using precast and ready-mix concrete. The GaBi construction materials database is used altogether with openLCA software, which enhances significantly quantifying environmental footprints of constructions. Results are analyzed at the construction material level to enhance sustainability aspects within design of constructions. Higher environmental impact of UHPC is shown when only the quantitative aspects are considered for the formulation of the functional unit. At the case study level, results are analyzed to test the footprints of UHPC in relation to the functionality provided. Better environmental performance of UHPC could be indicated within the assessment of a practical application such as a bridge. Cumulative raw material demand in terms of RMI and total primary material extraction in terms of TMR, in addition to water use of UHPC, are provided for the first time by this article. UHPC shows relative lower footprint values when the use phase is considered. However, higher environmental impacts are shown within the end-of-life phase according to the high compressive strength. Policies and standards related to UHPC use need to be developed according to the current increase in the usage of new concrete technologies. Because of the relative high cost of UHPC, an economic analysis, e.g., life cycle costing of UHPC in comparison with CC, could be a crucial issue to be considered for future studies.

## Figures and Tables

**Figure 1 materials-12-00851-f001:**
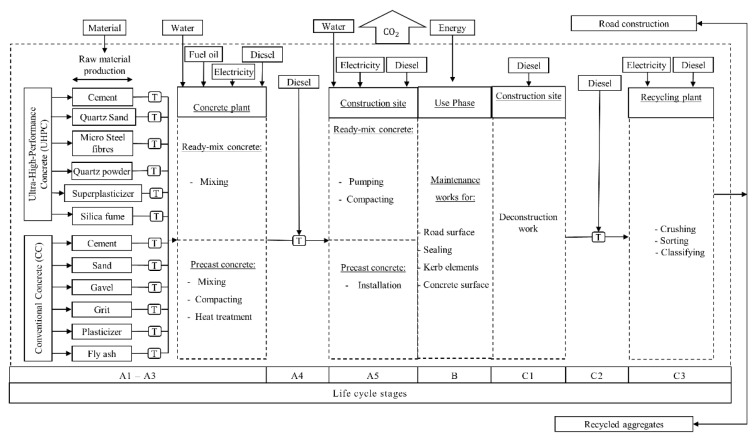
Diagram of life cycle assessment processes of cradle-to-grave of Ultra-High-Performance Concrete (UHPC) and Conventional Concrete (CC). Note: T = Transport.

**Figure 2 materials-12-00851-f002:**
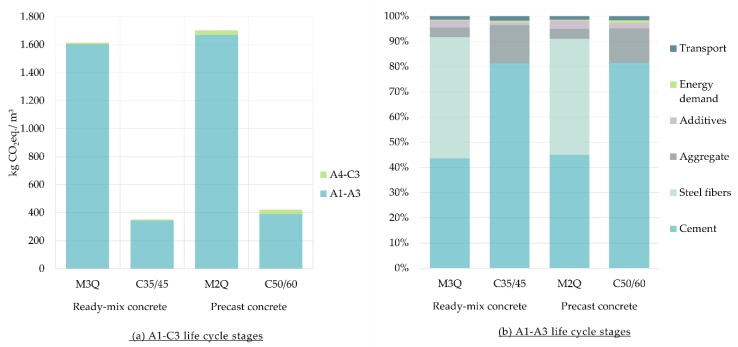
(**a**) Carbon Footprint of Ultra-High-Performance Concrete (M3Q and M2Q) in comparison with conventional concrete (C35/45 and C50/60) per m^3^; (**b**) Share of materials of the carbon footprint for the A1–A3 life cycle stages.

**Figure 3 materials-12-00851-f003:**
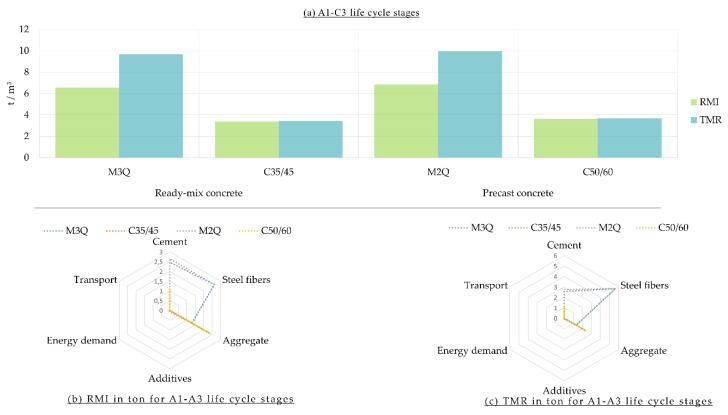
(**a**) Material Footprint of Ultra-High-Performance Concrete (M3Q and M2Q) in comparison with CC (C35/45 and C50/60) per m^3^; (**b**) Share of materials of the Raw Material Input (RMI) for the A1–A3 life cycle stages; (**c**) Share of materials of the Total Material Requirement (TMR) for the A1–A3 life cycle stages.

**Figure 4 materials-12-00851-f004:**
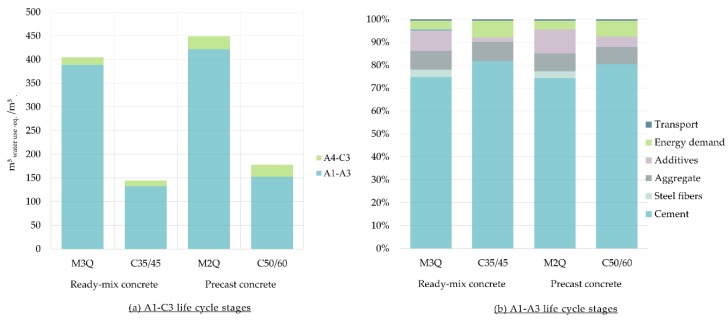
(**a**) Water footprint of Ultra-High-Performance Concrete (M3Q and M2Q) in comparison with conventional concrete (C35/45 and C50/60) per m^3^; (**b**) Share of materials of the water footprint for the A1–A3 life cycle stages.

**Figure 5 materials-12-00851-f005:**
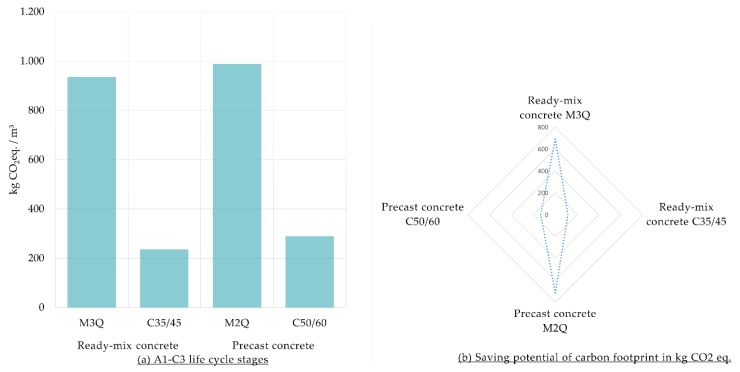
(**a**) Carbon Footprint of the Ultra-High-Performance Concrete (M3Q and M2Q) in comparison with conventional concrete (C35/45 and C50/60) regarding sensitivity analysis per m^3^; (**b**) Potential savings of carbon footprint in kg CO_2_ equivalents regarding sensitivity analysis.

**Figure 6 materials-12-00851-f006:**
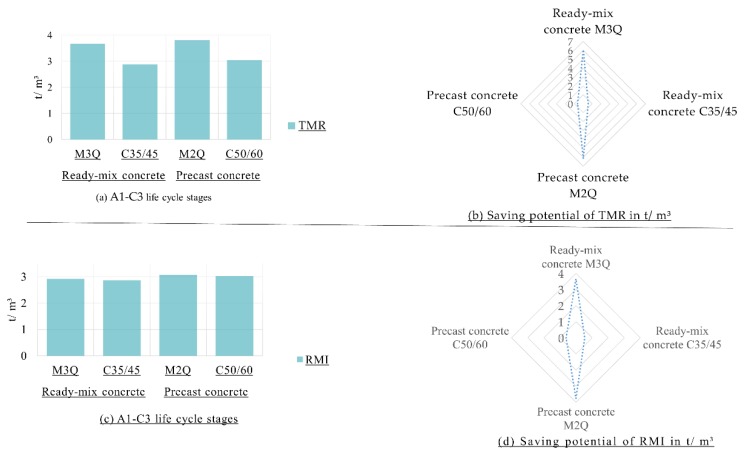
(**a**) Material Footprint in terms of TMR of the Ultra-High-Performance Concrete (M3Q and M2Q) in comparison with conventional concrete (C35/45 and C50/60) regarding sensitivity analysis; (**b**) Potential savings of TMR in ton regarding sensitivity analysis; (**c**) Material Footprint in terms of RMI of the Ultra-High-Performance Concrete (M3Q and M2Q) in comparison with conventional concrete (C35/45 and C50/60) regarding sensitivity analysis; (**d**) Potential savings of RMI in ton regarding sensitivity analysis.

**Figure 7 materials-12-00851-f007:**
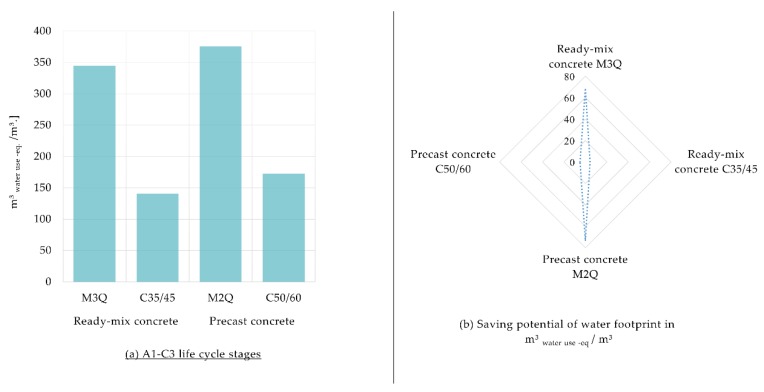
(**a**) Water Footprint of the Ultra-High-Performance Concrete (M3Q and M2Q) in comparison with conventional concrete (C35/45 and C50/60) regarding sensitivity analysis; (**b**) Potential savings of water footprint in m^3^ water use eq. per m^3^ regarding sensitivity analysis.

**Figure 8 materials-12-00851-f008:**
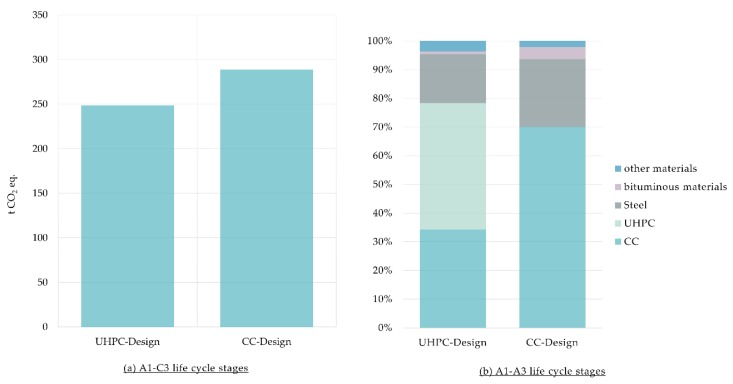
(**a**) Carbon footprint of the two designs of the bridge, Conventional Concrete Design (CC-Design) and Conventional Concrete enhanced with Ultra-High-Performance Concrete (UHPC- Design); (**b**) Share of materials of the carbon footprint for the A1–A3 life cycle stages.

**Figure 9 materials-12-00851-f009:**
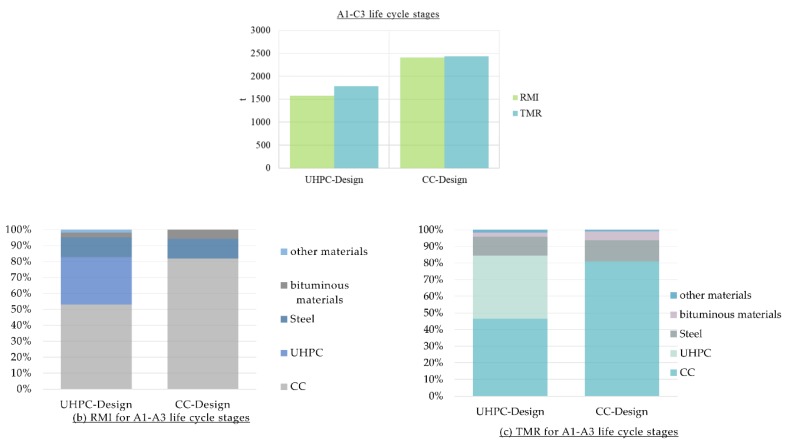
(**a**) Material footprint of the two designs of the bridge, Conventional Concrete Design (CC-Design) and Conventional Concrete enhanced with Ultra-High-Performance Concrete (UHPC-Design); (**b**) Share of materials of the RMI for the A1–A3 life cycle stages; (**c**) Share of materials of the TMR for the A1–A3 life cycle stages.

**Figure 10 materials-12-00851-f010:**
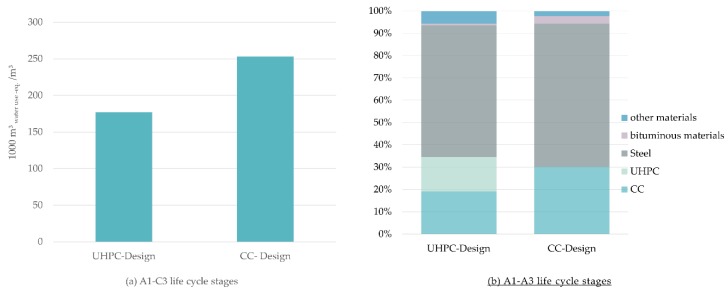
(**a**) Water footprint of the two designs of the bridge, Conventional Concrete Design (CC-Design) and Conventional Concrete enhanced with Ultra-High-Performance Concrete (UHPC-Design); (**b**) Share of materials of the water footprint for the A1–A3 life cycle stages.

**Table 1 materials-12-00851-t001:** Reference mixtures of Conventional Concrete (CC) and Ultra-High-Performance Concrete (UHPC) [14,40].

Material	Unit	UHPC		CC	
		Ready-mix	Precast	Ready-mix	Precast
		M3Q	M2Q	C35/45	C50/60
Cement/CEM I 52.5 R-HS/NA (SR 3)	kg/m^3^	775	832	-	-
Cement/CEM II/A	kg/m^3^	-	-	356	405
Quartz sand	kg/m^3^	946	975	-	-
Sand	kg/m^3^	-	-	640	654
Gravel	kg/m^3^	-	-	806	827
Grit	kg/m^3^	-	-	362	371
Water	kg/m^3^	183	166	165	141
Quartz powder	kg/m^3^	193	207	-	-
Silica fume	kg/m^3^	164	135	-	-
Fly ash	kg/m^3^	-	-	47	25
Superplasticizer	kg/m^3^	23.50	29.40	-	-
Plasticizer	kg/m^3^	-	-	1.80	4.60
Micro steel fibers	kg/m^3^	192	192	-	-
Total	kg/m^3^	2476.50	2536.40	2377.80	2427.60

**Table 2 materials-12-00851-t002:** Transportation types and distances of concrete ingredients.

Material	Transport Distance (km)	Type of Transport			Reference
		Truck %	Train %	Ship %	
Sand; Gravel; Grit	39	88	1.90	10.10	[45]
Cement	106	79.90	8.90	11.20	[45]
Fly ash	100	100	-	-	[45]
Plasticizers	100	100	-	-	[45]
Quartz sand; Quartz powder	150	100	-	-	Interviews
Silica fume	700	100	-	-	Interviews
Steel fibers	230	100	-	-	Interviews

**Table 3 materials-12-00851-t003:** Inventories of the required energy regarding A3 life cycle stage of the production of 1 m^3^ concrete.

Energy Carrier	UHPC		CC	
	Ready-mix	Precast	Ready-mix	Precast
	M3Q	M2Q	C35/45	C50/60
mixing and plant operation				
Electricity (kWh/m^3^)	7.09		4.43	
Fuel oil (L/m^3^)	0.26			
Diesel	0.09			
Compacting of precast concrete				
Electricity (kWh/m^3^)	-	0.90	-	0.90
Heat treatment of precast concrete				
Fuel oil (L/m^3^)	-	3.00	-	

**Table 4 materials-12-00851-t004:** Construction materials of the two design options of the bridge.

Bridge Structure Element	UHPC Design	Quantity	CC-Design	Quantity
Abutments and middle support with 5 single piers	C35/45	25.00 m^3^	C35/45	40.00 m^3^
	Reinforcing steel	6.50 t	Reinforcing steel	27.00 t
Support over the 5 middle single piers	Concrete hinges with UHPC cast on site (M2Q)	1.10 m^3^	-	-
Main girders (longitudinal)	Precast concrete C50/60	180.00 m^3^	Precast concrete C50/60	142.00 m^3^
	Reinforcing + Prestressing steel	54.80 t	Reinforcing + Prestressing steel	33.00 t
	Polyurethane putty	0.12 t	C35/45	119.00 m^3^
	M3Q	7.00 m^3^	-	-
Sealing	M3Q	20.50 m^3^	Epoxy resin	0.56 t
	-	-	Quartz sand	0.84 t
	-	-	Bitumen sheets	3.09 t
Road surface	Poured asphalt	1.73 t	Poured asphalt	56.45 t
	Stone mastic asphalt (SMA)	28.80 t	-	-
	Jointing material	0.09 t	Jointing material	0.16 t
	High-grade chippings (0/5)	0.90 t	Aggregates (2/5), (5/8)	1.95 t
Kerb elements	Epoxy resin	1.24 t	-	-
	M2Q	33.00 m^3^	Ready-mix concrete C25/30	59.00 m^3^

**Table 5 materials-12-00851-t005:** Frequency of replacements required for the maintenance work during use phase of the bridge.

Element	Renewal	
	UHPC Design (Number)	CC-Design (Number)
Road surface	4	4
Sealing	-	4
Kerb elements	-	2
Concrete surface	-	2

**Table 6 materials-12-00851-t006:** Types of cement of the considered mixtures of concrete in terms of results of the sensitivity analysis.

Type of Concrete		Quantity (kg/m^3^)	Ex-Ante Type of Cement	Replaced by
Ready-mix concrete	UHPC M3Q	775	CEM I 52.5	CEM III/A
	C35/45	356	CEM II/A	
Precast concrete	UHPC M2Q	832	CEM I 52.5	
	C50/60	405	CEM II/A	

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
