# Peer review of "Environmental Assessment of Ultra-High-Performance Concrete Using Carbon, Material, and Water Footprint"

_materials, 2019, doi:10.3390/ma12060851_

Round 1
Reviewer 1 Report
Ms. Ref. No.: materials-457393-peer-review-v1- Reviewer 1 comments
Environmental Assessment of Ultra-High-Performance Concrete using Carbon, Material, and Water Footprint
Reviewer comments:
SUMMARY
The manuscript deals with the use of Life Cycle Assessment (LCA) for Ultra-High-Performance Concrete (UHPC) in comparison with Conventional Concrete (CC), in terms of carbon, material, and water footprint. Environmental impacts are determined for the cradle-to-grave life cycle of the UHPC, considering precast and ready-mix concrete. This is a topic that has not been widely covered in the literature, therefore, this a subject of great interest, but it is somehow limited in the analysis and application of these results.
MAIN IMPRESSIONS
This paper has an undeniable practical usefulness. However, from a scientific point of view, the following issues must be addressed: i) The type of Cement has a significant influence in the LCA, therefore, it must be clearly defined in 2.1; ii) Ultra-High-Performance Concrete (UHPC) should be compared with other similar concretes such as BPR, Ductal, CRC and so on, and iii)Many of the 67 references belong to the German literature, it is recommended to extend the discussion of the present results to the worldwide literature.
MORE DETAILED COMMENTS
Introduction & Discussion: Ultra-High-Performance Concrete (UHPC) should be compared with other similar concretes such as:
BPR: N. Roux; C. Andrade; M.A. Sanjuán. Experimental Study of Reactive Powder Concretes (RPC). Journal of Materials in Civil Engineering A.S.C.E. 1996; 8(1) 1-6. http://dx.doi.org/10.1061/(ASCE)0899-1561(1996)8:1(1))
Ductal: Mohamadreza Shafieifar, Mahsa Farzad, Atorod Azizinamini. Experimental and numerical study on mechanical properties of Ultra High Performance Concrete (UHPC). Construction and Building Materials, Volume 156, 15 December 2017, Pages 402-411. https://doi.org/10.1016/j.conbuildmat.2017.08.170
CRC: Tommy Bæk Hansen and Bendt Aarup. "Engineering challenges in international application of UHPFRC" , publiced in Concrete Engineering International, 2015, vol. 19, Issue 1. http://www.crc-tech.com/Admin/Public/DWSDownload.aspx?File=%2fFiles%2fFiler%2fhicon%2fnyheder+og+presse%2fCEI+V19I01P13.pdf
Page 3: Cement designations must be corrected according to EN 197-1:2011. For instance, CEM I 52.5 should be R or N. With regard to cement CEM II/A, there are 19 subtypes.
Reference [37] should be checked. Probably, prEN 197-1:2018 will be withdrawn in March 2019. It is recommended to reference EN 197-1:2011.
In 3.2.1 is mentioned CEM I 52.5 R-HS / NA. According to EN 197-1:2011, a sulfate resistance cement must be named as SR 0, 3 or 5. The low alkali content is not standarised currently in Europe. Then, could you please add the chemical composition of this cement in 2.1.
Please check the type of cement CEM III 52.5 cited in [66]. Have you taken into account the CO2 allocation corresponding to the blast-furnace slag of this cemet type?
Conclusion is short and clear: ”Better environmental performance of UHPC could be indicated within the assessment of a practical application such as a bridge”. After disscussing the potential use of other types of UHPC such as BPR, Ductal and CRC, will you get the same conclusion?
Conclusion: Please add some conclusions about the GaBi database. In your opinion is a good tool?
In general, the numbers and letters of the Figures should be bigger.
RECOMMENDATION
In conclusion, Minor changes have been proposed.
Author Response
Response to Reviewer 1 Comments
Point 1: English language and style are fine/minor spell check required.
Response 1: Thanks, spelling is reviewed and rechecked.
Point 2: Does the introduction provide sufficient background and include all relevant references? Must be improved
Response 2: Thanks, more literature about UHPC are considered in the introduction section.
Point 3: Ultra-High-Performance Concrete (UHPC) should be compared with other similar concretes such as BPR, Ductal, CRC and so on,
Response 3: Thanks, This point is declared in many points within the paper such as in section 3.4 (Summary of analyses at the material and case study. level), and in section 4. (conclusion)
Point 4: Many of the 67 references belong to the German literature, it is recommended to extend the discussion of the present results to the worldwide literature. Introduction & Discussion: Ultra-High-Performance Concrete (UHPC) should be compared with other similar concretes such as:
Response 4: Thank you very much, 4 worldwide literature reviewed and cited in the introduction.
Point 5: Page 3: Cement designations must be corrected according to EN 197-1:2011. For instance, CEM I 52.5 should be R or N. With regard to cement CEM II/A, there are 19 subtypes.
Response 5: Thank you very much, Cement designations are corrected according to EN 197-1:2011, CEM I 52.5 R-HS / NA (SR 3) and the average of cement CEM II/A is considered.
Point 6: Reference [37] should be checked. Probably, prEN 197-1:2018 will be withdrawn in March 2019. It is recommended to reference EN 197-1:2011.
Response 6: Thank you very much, done.
Point 7: In 3.2.1 is mentioned CEM I 52.5 R-HS / NA. According to EN 197-1:2011, a sulfate resistance cement must be named as SR 0, 3 or 5. The low alkali content is not standarised currently in Europe. Then, could you please add the chemical composition of this cement in 2.1.
Response 7: thank you, CEM I 52.5 is SR 3. Which is added in section 2.1-
Point 8: Please check the type of cement CEM III 52.5 cited in [66]. Have you taken into account the CO2 allocation corresponding to the blast-furnace slag of this cement type?
Response 8: CO2 allocation corresponding to the blast-furnace slag is included within the life cycle inventory of CEM III 52.5.
Point 9: Conclusion is short and clear: ”Better environmental performance of UHPC could be indicated within the assessment of a practical application such as a bridge”. After discussing the potential use of other types of UHPC such as BPR, Ductal and CRC, will you get the same conclusion?
Response 9: UHPC is compared with CC at case study level considering the final functionality. This issue is further discussed within the section 3.4 (summary of analyses at material and case study level) and section 4 (conclusion)
Point 10: Conclusion: Please add some conclusions about the GaBi database. In your opinion is a good tool?
Response 10: thank you, done.
Point 11: the numbers and letters of the Figures should be bigger.
Response 11: Thank you very much, done.
Reviewer 2 Report
The manuscript deals with an environmental evaluation of two concrete types, CC and UHPC, regarding their impacts to global warming, as well as analysing the water and material consumption. The authors analyse the environmental impacts using Gabi software. I appreciate the verification of the results using the sensitivity analysis. The manuscript is written well, the methodology is appropriate and the input parameters are described. In spite of that, I have several question to precise the inputs:
Fig. 1 – the authors declare that the evaluation of the materials was performed within the cradle-to-grave boundaries, however, the use phase (operation of the bridge) is not clearly illustrated even characterised in the Fig. 1.
When analysing the whole life cycle of the bridge, what percentage of material recycling was considered?
Were fly ash and silica fume considered as secondary waste materials? Were their impacts counted to be equal zero or were there calculated any impacts related to their treatment before incorporating them into concrete?
Please check the text again, there are several formal mistakes in the manuscript, e.g. small letters at the beginning the sentences etc.
Author Response
Response to Reviewer 2 Comments
Point 1: English language and style are fine/minor spell check required.
Response 1: Thanks, spelling is reviewed and rechecked.
Point 2: Fig. 1 – the authors declare that the evaluation of the materials was performed within the cradle-to-grave boundaries, however, the use phase (operation of the bridge) is not clearly illustrated even characterized in the Fig. 1.
Response 2: Thanks, the figure is rewritten. Use phase of the bridge is described.
Point 3: When analyzing the whole life cycle of the bridge, what percentage of material recycling was considered?
Response 3: Thanks, the percentage of material recycling is similar to that considered within the Life Cycle Inventory (LCI) of the 1m³ of UHPC and CC.
Point 4: Were fly ash and silica fume considered as secondary waste materials? Were their impacts counted to be equal zero or were there calculated any impacts related to their treatment before incorporating them into concrete?
Response 4: Thank you very much, this sentence is added section 2.2: Secondary materials are allocated according to DIN EN 15804 [32]. Fly ash is a by-product of the coal power station. Silica fume is a by-product of the production of silicon metal or ferrosilicon alloys. Therefore, their environmental impacts are only calculated for transport to the concrete plant.
Point 5: Please check the text again…
Response 5: Thank you, done.
